# Porous Coatings to Control Release Rates of Essential Oils to Generate an Atmosphere with Botanical Actives

**DOI:** 10.3390/ma15062155

**Published:** 2022-03-15

**Authors:** Kai Hettmann, Fabien W. Monnard, Gabriela Melo Rodriguez, Florentine M. Hilty, Selçuk Yildirim, Joachim Schoelkopf

**Affiliations:** 1Omya International AG, Baslerstrasse, 4665 Oftringen, Switzerland; fabien.monnard@omya.com (F.W.M.); gabriela.melorodriguez@omya.com (G.M.R.); florentine.hiltyvancura@omya.com (F.M.H.); 2Life Sciences and Facility Management, Zurich University of Applied Sciences, Campus Reidbach, 8820 Waedenswil, Switzerland; yise@zhaw.ch

**Keywords:** antimicrobial, evaporation, layers, linear driving force, packaging

## Abstract

Essential oils have been used in diverse areas such as packaging, agriculture and cosmetics, for their antimicrobial and pesticide activity. The organic volatile compounds of the essential oils are involved in its activity. Controlling their release helps to prolong their functionality. In this study, a functionalized calcium carbonate porous coating was employed to control the release of thyme and rosemary oil in a confined space. The release rate was evaluated at 7 °C and 23 °C, gravimetrically. It was shown that the capillary effect of the porous coating slowed down the release of the volatiles into the headspace compared to the bulk essential oil. A linear drive force model was used to fit the obtained data from both essential oils. The model showed that rosemary reached the asymptotic mass loss equilibrium faster than thyme. This result can be explained by the diverse composition and concentration of monoterpenoids between the two essential oils. Temperature and degree of loading also played important roles in the desorption of the essential oils. It was observed that at high degrees of loading and temperatures the desorption of essential oils was higher. The above-described technology could be used for applications related to food preservation, pest control among others.

## 1. Introduction

Essential oils (EOs) have recently seen a boom in interest, mainly because of their antimicrobial properties [1,2,3] and their sustainable image as botanicals [4]. Their applications span many markets; from agricultural application for pests control [5,6], to animal feed and veterinary applications [7], to human medicine mainly in the field of dermatologic treatments [8], to use in functional packaging [9,10,11,12,13]. The latter allows for food waste reduction [14] and control of food spoilage by pathogenic microorganisms [15,16].

In dealing with EOs, there are many challenges, including: Their varying composition based on growing conditions and cultivar diversity with a multitude of single components [17], many of them being monoterpenoids [18], as in EO of rosemary and thyme reported in Table 1 [19]. A comparison of the properties of the EOs with those of the single constituents reveals that some components dissolve others which would be solid at room temperature (RT), and that azeotrope and eutectic effects play a crucial role [20,21].

Porous systems are extensively used as carriers for active substances [22,23,24,25,26]. In the case of liquids, the benefit is that the liquid is enclosed and no longer mobile. This may lead to easier and safer handling and dosing. Depending on the liquid saturation of the porous carrier, the liquid/vapor interface area is multiplied and may lead to faster evaporation rates in early stages and slower rates in later stages.

Different researchers have applied the concept of combining EO with some form of support, be it in the form of encapsulation or compounding into polymers or gels [27,28]. There are few examples of using porous minerals as carriers [29,30,31,32].

In this work, porous layers were achieved by formulating highly porous functionalized calcium carbonate (FCC) mineral particles with an adequate binder system. This coating was applied to polymeric planar substrates. The binder type and level were set to give an open, interconnected pore structure during drying. This, in turn, enabled loading and controlled release of EO. Loading was achieved by a spray technique. The sum release rate into a closed air volume was measured by the means of an Intelligent Gravimetric Analyzer (IGA). Furthermore, the dynamics of the single constituents were analyzed using mass spectroscopic (MS) approach. To the knowledge of the authors, this is the first time quantitative data of this kind has been presented in a scientific paper.

Our motivation to analyze release rates of EO loaded into FCC comes from the close work with the Yildirim group [19], which used EO to create an antimicrobial atmosphere in sealed food packaging. Past work tested botanicals such as clove oil, rosemary oil, thyme oil, cinnamon oil, and oregano oil for their antimicrobial activity. In this work, thyme oil and rosemary oil have been chosen due to their acceptable stability on said FCC particles and superior performance in controlling microbial activity on meat in closed packaging.

## 2. Materials and Methods

### 2.1. Chemicals

EOs, from rosemary and thyme, namely Rosmarinöl natürlich rekt. EuAB/01-5600 and Thymianöl rekt. EuAB/01-6650 (trade names), from Hänseler (Herisau, Switzerland), were used. EOs were stored in the dark at 21 ± 1 °C and utilized before the expiration date.

The FCC particles, provided by Omya international AG, contain a lamellae layer of 47% of calcium phosphate and the remaining 53% is composed of calcium carbonate. The FCC used had a median particle size (d50) of 5.5 μm, a specific surface area of 110 m^2^/g and an intraparticle pore volume of 1.22 cm^3^/g as measured by mercury intrusion porosimetry (MIP).

Sodium-neutralized polyacrylate dispersing agent was provided by Coatex (Genay, France) and the Acronal 500D acrylic latex binder was provided BASF (Ludwigshafen, Germany). Other chemicals were purchased from Sigma Aldrich at ACS Reagent Grade.

### 2.2. Preparation of the FCC Film

A sodium-neutralized polyacrylate dispersing agent (9.8 g, 42% solid content) was dispersed in water (284 mL) and FCC (205.8 g) was added step by step until a homogeneous formulation was obtained. Polyacrylate binder Acronal 500D (45.6 g, 46 wt.% solid content) was adjusted to pH 9 with NaOH 30 wt.% and added to the previous solution to obtain the coating formulation. Before use, the coating composition was stirred for 5 min to achieve a homogeneous distribution of the mineral in the coating formulation. Finally, the coating compound was applied using a coating table RK303 multicoater (Erichsen) onto a PET film Hostaphan RN 100 µm (PützFolie, Germany). Drying was done using infrared heating and hot air impingement with a S-Dryer machine (Durrer, Switzerland). The coat weight target was 50 g/m^2^. The final coating porous structure was inspected by SEM microscopy (Figure 1).

### 2.3. Loading of EO on PET Foils

PET films coated with 50 g/m^2^ of previously described FCC coating were sprayed with EO. The spray system used to load the EO was a 3-Axis E2 series automated fluid dispensing robot with a low-pressure spray valve (781 mini 0.76 mm) from Nordson. Three different loading amounts of EO (5, 10 and 30 *w*/*w*%, relative to the coating mass) were sprayed on the coated films and immediately gravimetrically verified by a microbalance. The freshly EO loaded samples were immediately transferred to closed containers and brought to the analytical devices for instant measurements to minimize any loss of EO’s volatiles.

In the spray device, some parameters were maintained constant such as: 1. the material pressure at 0.3 bars and 2. the air pressure at 4 bars. Other parameters were varied to obtain the desired loading amount of the EO on the coating: 1. the valve opening from the spray nozzle was graduated between 5 to 12 units and 2. the line speed was changed between 19 to 150 mm/s. As described above, the freshly sprayed coated films were transferred to the Intelligent Gravimetric Analysis (IGA) (20 × 12 mm) and mass spectrometer (30 × 30 mm), for further characterization.

### 2.4. Release Experiments Using Gravimetrical Analysis

Experiments on the release of EO from coatings were done by gravimetrical monitoring of the EO content of the samples. The test unit was a Hiden Isochema IGA-002, which provides an environment for controlled conditions during the experiments. The sample was mounted in the reaction chamber and the chamber was immediately closed afterwards. The temperature of the sample chamber was controlled by a thermostat and kept at 7 ± 1 °C or 23 ± 1 °C, during the entire measurement. The time from loading of EO on the PET foils containing FCC coating to start of the release experiment was kept as short as possible (less than 10 min).

The experiment started with a purging procedure. The chamber was pressurized using 99.99% nitrogen (grade 4.0) at 200 mbar/min until the pressure was reached to 8 bar. Afterwards, the chamber was depressurized at the same rate down to atmospheric pressure (1013 ± 10 mbar). This procedure was repeated two times, in- and outlet of the chamber was closed after that. Then, the experimental clock and weight were set to zero and the gravimetrical change was recorded for 20 h with a resolution of 100 ng. The recorded data was used to fit a linear driving force (LDF) model [33].

### 2.5. Release Experiments Using Mass Spectrometry

In addition to the gravimetric analysis, the release of EO was also analyzed by mass spectrometry. This analysis was conducted by loading the sample sheet of FCC-EO coated PET film into a 200 mL PTFE reactor with a controlled temperature at 7 °C or 23 °C. The reaction chamber was closed immediately after loading and remained closed except for a connection to the heated inlet capillary of a MKS Cirrus 3XD mass spectrometer, which was opened for 30 min measurements after 2 h and 20 h.

The mass spectrometer was calibrated for mass by using 5.0 He, 4.0 Kr and 4.0 Xe gas before and after the experimental series. No significant shift in mass could be observed. During the experiments, the mass range from 100 amu to 160 amu was scanned continuously at a resolution of 1/64 amu for 15 s. The outlet of the mass spectrometer was reinjected into the sample chamber for constant pressure conditions. The mass spectrometer was purged using 5.0 He gas for the complete time not connected to the sample chamber.

## 3. Results

### 3.1. Theoretical Considerations

If a liquid is absorbed, often called “loaded”, into a mesoporous network, a number of complex interactions start to take place. In most cases, the loading is driven by capillary forces defined by curvature of the liquid/vapor interfaces formed in the ganglia of the network. The dynamics of this process, called imbibition or incipient impregnation, have been described in detail and are still today the subject of various modelling approaches.

The release of liquid vapor from an oversaturated porous structure into an infinite air volume may be divided into 4 stages:

Oversaturation means the surface of the porous substrate, in our case, the coating, is surface-wet and liquid evaporates until the surface is dry. This transition may be observed along with a well-visible change in optical appearance from reflective gloss to a matte aspect.
(1)$$\frac{\ln P_{K}}{P_{V}}=\frac{2\gamma_{LV}\cos\theta }{r_{c}RT}<0\;$$

The macroscopic process of desorption is governed by many mechanisms on the pore scale level. Liquid/vapor menisci form at the pore entrances and evaporation continues from these menisci. The menisci change the vapor pressure of the liquid due to the Kelvin–Tompson effect Equation (1). Into a slower evaporating system.

Where *R* is the gas constant, *T* the temperature, *P_v_* is the vapor pressure over a flat liquid vapor interface. *P_k_* is the Kelvin pressure, which is lower than *P_v_* for a concave liquid phase (capillary) and higher for a convex one (droplet). *γ* is the liquid vapor interfacial tension, *θ* is the contact angle and *r_c_* is the radius of the capillary with circular cross section

Menisci retreat into the structure. Liquid diffuses through the empty ganglia and re-condenses on the pore walls based on the momentary equilibrium conditions. Every meniscus in the structure exerts a Laplace pressure inversely linear to the pore diameter.
(2)$$\Delta P=\frac{\gamma_{LV}\cos\theta }{r_{c}}$$
where *ΔP* is the pressure difference across the curved liquid vapor interface, i.e., the meniscus. Therefore, larger pores empty first, while smaller pores retain liquid.

As liquid connectivity in the pore network ruptures, the competition of the menisci for liquid stalls and menisci eventually collapses into liquid films wetting the pore walls. At this stage, the liquid continues to evaporate to the point of adsorbed layers of liquid molecules in equilibrium with the atmosphere.

If the liquid desorbs into a confined space, the saturation of the vapor in the confined volume comes into play. At the end of the process, a liquid distribution equilibrium is established between the confined space and the porous structure, yielding an equilibrium mass, desorbed at infinite time, also called the asymptotic mass loss, as measured in our experimentation.

While above equations describe pore level phenomena, the overall diffusion process is typically described by relating the mass flux to the gradient of concentration over distance, as described in Fick’s laws. For diffusion in porous media, Webb and Pruess [34] identified the need for taking molecule-pore wall interactions into account comparing the advective diffusive model with the dusty gas model as implements in Fick’s equation. However, both expressions are rather complicated compared with a much simpler model. The LDF model stems from Glueckauf and Coates [35], who derived the approximation relating the average adsorbate concentration inside a porous particle directly with the concentration in the fluid phase.

Sircar and Hufton [36] give a thorough explanation and analysis as to why the LDF model works. Their conclusion:

The Fickian Diffusion model, which is a special case of the most rigorous chemical potential driving force (CPDF) model of fluid transport within a porous, adsorbent particle, is often used for analyzing isothermal gas or vapor uptake by porous substrates in order to estimate a diffusivity parameter.

The linear driving force (LDF) model with a lumped mass transfer coefficient, on the other hand, is very frequently used for practical analysis of particle bed dynamic data and for adsorptive process design [37,38] because it is simple, analytical, and physically consistent.

$$\begin{array}{cc} \mathrm{Model}\;\mathrm{Constant} & \;P\;\mathrm{experiment}\\ \mathrm{Fickian}\;\mathrm{diffusion} & \;f\left(t\right)=\overset{\infty }{\underset{n=1}{\sum }}\frac{6}{{\left(n\pi \right)}^{2}}\;e^{-\left[Dn^{2}\pi^{2}t/R^{2}\right]}\\ \mathrm{LDF} & \;f\left(t\right)=1-e^{-kt} \end{array}$$
where *n* is the fluid loading in the particle, *R* is the particle diameter, *D* is Fick’s diffusivity, while *k* is the mass transfer coefficient. For the modeling of the experimental data in this paper, the LDF model is applied by fitting the experimental results to the equation
(3)$$m\left(t\right)=\Delta m\times \left(1-e^{-\frac{t}{k}}\;\right),$$
with *m* as the change in mass of the sample at time *t*, Δ*m* as the equilibrium mass desorbed at infinite time, also called the asymptotic mass loss, and *k* as the mass transfer coefficient. The LDF model can be used to separate the kinetic parameter from the equilibrium adsorption in a generalized way without explicitly modeling separate physical phenomena such as small-scale heterogeneities, coating roughness and other effects.

### 3.2. Release of EO

The gravimetrical change of the sample is a measure of the release of an EO [39]. The decrease in weight can be directly correlated to the mass of EO released into the atmosphere. The release of thyme and rosemary oil are shown in Figure 2a,b, respectively), the trend for both cases are not linear. It is observed that the amount released is generally dependent on the temperature. The type of EO, which is highly related with the chemical composition of each EO, and the degree of loading, are also important parameters.

At higher temperatures, more EO is released. This is expected due to the dependence of equilibrium vapor pressures on temperature. Figure 3 shows data from single components measured via GC MS. These exhibited the above-mentioned behavior [40,41,42].

At longer timescales, higher loadings resulted in higher mass release and lower loadings had lower mass released. The dependence of the release on the degree of loading is more complex at shorter time scales (less than 1 h for thyme oil and less than 5 min for rosemary oil). To be able to separate the effects responsible for these observations, a more detailed physical model would need to be employed.

### 3.3. Equilibrium Vapor Pressure in Confined Space

The data obtained during the experiments was fitted to the LDF model (Fittings shown in support information). The asymptotic mass loss is a fitting parameter of this model. Since the experiments are done in a constant volume, the asymptotic mass loss is equivalent to the concentration of the substance in the gas phase at infinite time, and therefore the equilibrium vapor pressure in the confined space. This connection allows us to see differences of the equilibrium vapor pressure depending on temperature, EO composition, and the degree of loading. This differs from the equilibrium vapor pressure of the pure components of the EO, due to mixing energy, pore structure and temperature.

A comparison of the asymptotic mass loss of the EO not loaded into a porous structure to the concentrations expected for the pure compounds vapor pressures (Table 1) in a 2 L volume shows an interesting relationship. Note that there is a decrease of the equilibrium vapor pressure of the mixture compared to the summed partial vapor pressures of the pure substances, as may be seen in Figure 4. This is caused by the mixing energy involved in the EO, as the vapor pressure of a mixture directly depends on the mixing energy [43]. However, this relationship does not account for the differences in the asymptotic mass loss seen within the data of both EO, since the mixing energy does not change in this case.

The effect of temperature can be seen if the asymptotic mass loss of the EO is compared for each loading of an EO at both temperatures analyzed. Since the asymptotic mass loss of the samples analyzed at 7 °C and 23 °C increases with temperature, it can be assumed that the equilibrium vapor pressure does so as well. The increase is significantly larger for rosemary oil (60% on average) than for thyme oil (90% on average). Since such large differences are not known for the pure substances, this might be caused by differences in the structure of the liquid and associated mixing energies. However, it should also be noted that the variations in the data for rosemary oil is larger than for the thyme oil, and the scale of this effect is unknown. The variations between the two temperatures are smaller than the overall variation, showing that it is at least within normal storage temperatures and not the main controlling factor.

The effect of the pore structure can be shown if the various degrees of loading of the same EO are compared at constant temperature. In that case, different degrees of loading imply a different volume and shape of the filled pore space. A strong decrease in the asymptotic mass loss is observed from the bulk EO compared to the loaded, and a further decrease with less degree of loading. This implies that the equilibrium vapor pressure of the EO in the analyzed samples is mainly controlled by the pore structure. Associated properties like the Tomson Kelvin effect (Equation (1)), liquid distribution, as controlled by the pore level Laplace pressures (Equation (2)), and changes in the structure of the liquid phase [44] are also factors.

## 4. Discussion

### 4.1. Kinetics of EO Release

The LDF model also gives the mass transfer coefficient in addition to the asymptotic mass loss. The mass transfer coefficient describes the shape of the release curve in relation to time. Therefore, it provides information related to the kinetics of the release independent of the equilibrium vapor pressure. A small coefficient stands for a steeper gradient of desorption with abrupt transition towards equilibrium, and a larger one for a more sustained curve of release and a smoother transition into a final state. The kinetics are dependent on: (1) the condensation enthalpy [45], (2) the pore structure with the associated [46], (3) liquid/vapor interfacial mechanisms [47], (4) diffusion in gas and vapor phase, and (5) temperature.

The data derived from the experiments allows us to compare the kinetics in relation to temperature, EO composition and the degree of loading (Figure 5). The temperatures used cover the most important conditions for food storage in a fridge (7 °C) and at room temperature (23 °C). The relations of kinetics to the temperature differ between the two EO analyzed. A higher mass transfer coefficient implies a slower kinetic at higher temperature for the Thyme oil, if the total amount of released EO is not taken into account. For the Rosemary oil, the relationship is less clear, which might be caused by smaller overall variations observed. This also shows that the composition of the EO is one of the main factors for the control of the kinetics.

Since all other parameters are identical for the respective experiments, the differences between Thyme oil and Rosemary oil can only be explained by differences in composition. The effect of the pore structure can be seen if the experiments at different loadings are compared within the experiments for both EO. While there is a very strong increase in the mass transfer coefficient for increased degree of loading within the Thymol oil containing samples, the increase is less clear but observable for the Rosemary samples. This shows that depending on the EO composition, the effect of the pore structure on the release kinetics can be more or less pronounced. This is controlled by the interplay between heat and mass transfer [48], where the mass transfer is controlled by diffusion processes. The heat transfer is strongly influenced by heat capacity of the EO [49] and the activation energy of desorption controls the kinetic behavior (as shown for organic compounds in soil pores) [50].

### 4.2. Composition of the Released EO

The analysis of the vapor phase by mass spectrometry was used to determine the composition of the constituents in the released EO (Table 1). The data shows no systematic differences for the vapor phase composition within the analysis of each EO, but a very different composition between Thyme and Rosemary loaded samples (Figure 6). This relationship shows that the composition of the vapor phase is mainly controlled by the composition of the EO and not by the pore structure, temperature, or kinetics, at least within the limits of the experimental setup of measurements after 2 h and 20 h and 7 °C and 23 °C. Since there is no systematic difference between the data obtained after 2 h and 20 h, a strong fractionation change of the components during the release is not present (Figure 7).

### 4.3. Controlling the Release of EO

The insight into the controlling factors gained by the application of the linear driving force model can be used to control the release of an EO from a coating. In a closed system application such as food packaging, the concentration of the EO in the atmosphere can be controlled by the degree of loading into the porous coating. Higher degrees loadings lead to higher concentrations, and lower degrees of loading to lower concentrations. In contrast, lower degrees of loading show faster kinetics and can be applied if a fast release is desirable in the very early phase. In open systems both the kinetics and the equilibrium vapor pressure define the amount of EO released into the gas stream. In these cases, the concentration can be calculated from the linear driving force parameters (Figure 7), obtained for the specific EO and degree of loading using similar models as applied for gas filters in a reverse setting (e.g., Sircar 1983).

## 5. Conclusions

Porous coatings made of FCC are ideal carriers for EO to be released into vapor phase. This application could be more beneficial to use for confined spaces where the release of EO can be controlled such as packages, containers and bottles that present a sealing system.For both EO systems analyzed, rosemary and thyme, it was observed that rosemary reached faster the equilibrium of saturation in the vapor phase compared to thyme, which required almost double the time to be in the saturation point. A possible explanation for this effect is the different chemical composition of the EO. Instead of a liquid film, a dry surface is present on the coating, while the pore structure fixes the EO. The EO evaporates out of the pores and into the vapor phase and is distributed into the atmosphere.The release can be controlled by the degree of loading dependent on the EO used. This allows tailor-made solutions for a broad range of applications with control over the equilibrium concentrations and release kinetics.The results of this study also create new questions to be answered in future studies. The present study mainly focuses on the physical influences on the release, and the influence of the essential oil chemistry could not be resolved. More studies on the chemical composition of the vapor and the influence of the chemical composition of the essential oils on the release kinetics would complement the results of this study and complete the assessment of the controlling parameters.

## Figures and Tables

**Figure 1 materials-15-02155-f001:**
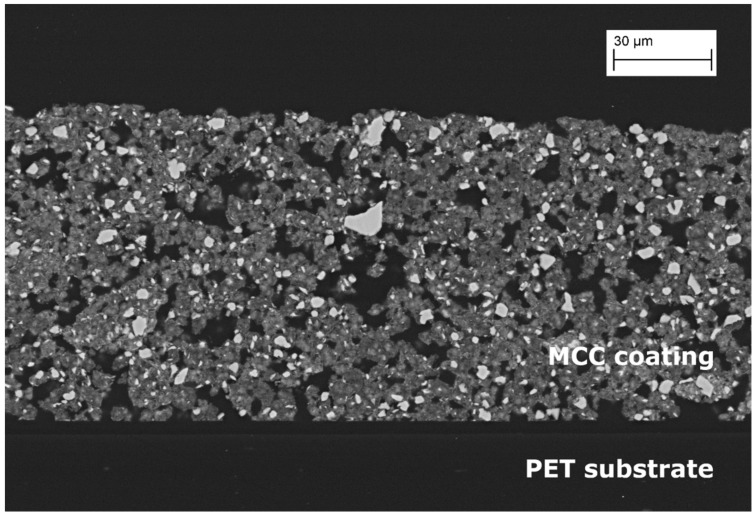
Cross section SEM image of the obtained FCC porous coating on PET, porous network.

**Figure 2 materials-15-02155-f002:**
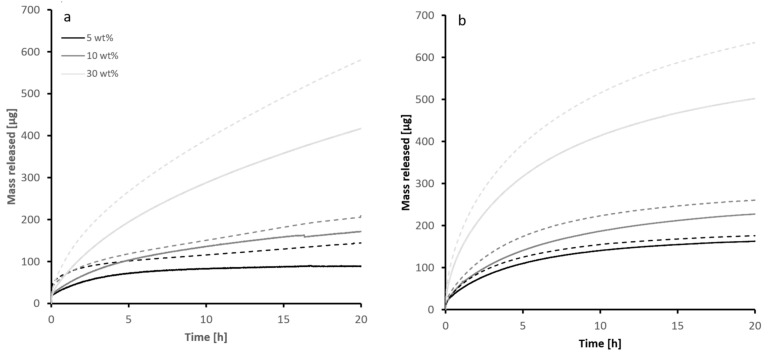
Experimental data for the release of thyme oil (**a**) and rosemary oil (**b**) over time. Grayscales represent different loadings.

**Figure 3 materials-15-02155-f003:**
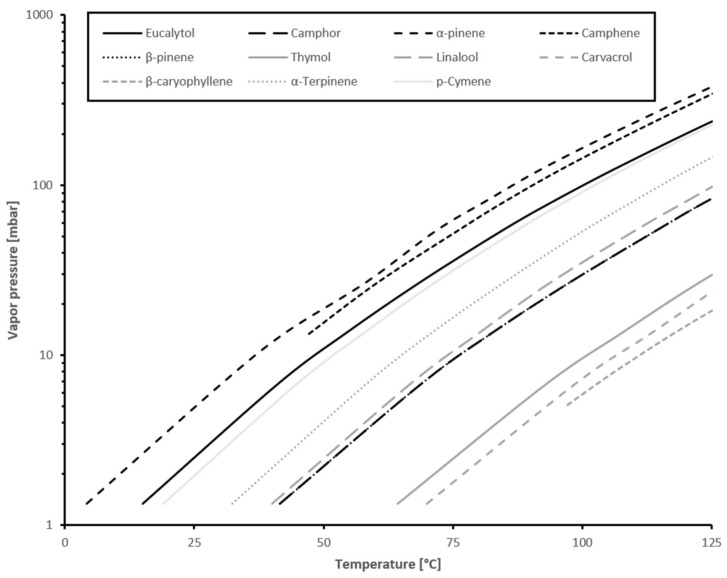
Vapor pressure of substances in this study. Data from Stull 1947, Perry et al., 2019 and Yao et al., 2019 [40,41,42].

**Figure 4 materials-15-02155-f004:**
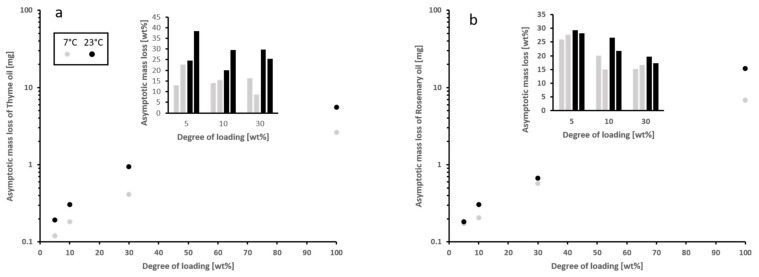
Asymptotic mass loss of EO (Thyme oil (**a**); Rosemary oil (**b**)) loaded onto coated samples and bulk shown at logarithmic scale. Inserts show the mass loss relative to the bulk EO.

**Figure 5 materials-15-02155-f005:**
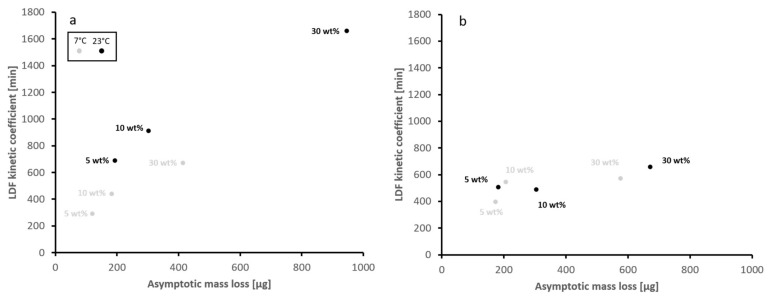
Linear driving force parameters deduced from the experiments for thyme oil (**a**) and rosemary oil (**b**). Every point indicates data for one degree of loading, as indicated in the figure.

**Figure 6 materials-15-02155-f006:**
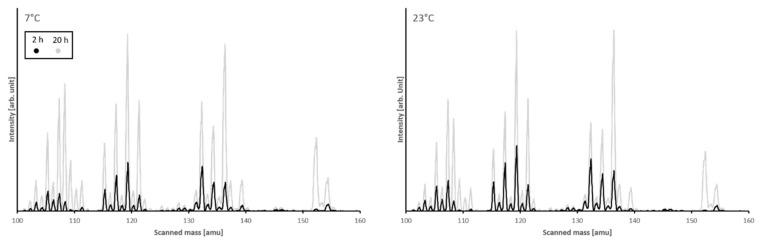
Mass spectra of experiments on 10 wt.% Rosemary loaded as an example.

**Figure 7 materials-15-02155-f007:**
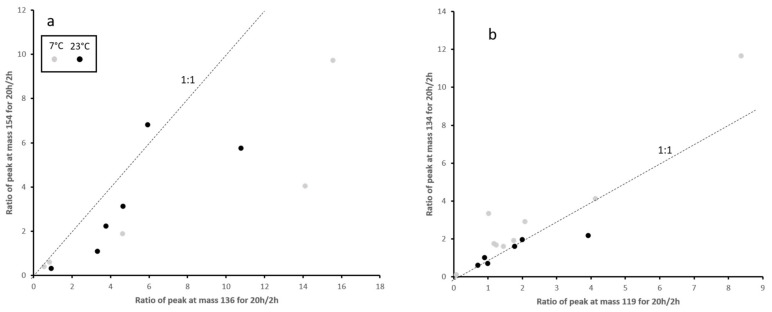
Ratios of mass spectroscopic intensity ratios for the time evolution of the peaks with the highest intensity of Thyme oil (**a**) and Rosemary oil (**b**) loaded samples.

**Table 1 materials-15-02155-t001:** Percentage content (integrated peak area) immediately after loading on FCC measured by extraction and GC-MS. Bold numbers indicate 5 most prevalent components in respective EO based on manufacturer data sheets.

Plant/Components	Rosmarinus Officinalis (Rosemary) [%]	Thymus Officinalis (Thyme) [%]
Eucalyptol	46.85 (±0.47)	0.80 (±0.05)
Camphor	16.79 (±0.24)	-
α-Pinene	9.49 (±0.33)	1.21 (±0.03)
Camphene	4.04 (±0.08)	0.94 (±0.02)
β-Pinene	4.26 (±0.12)	1.28 (±0.01)
Thymol	-	50.28 (±0.35)
p-Cymene	3.03 (±0.07)	21.73 (±0.07)
γ-Terpinene	0.10 (±0.01)	8.89 (±0.14)
Linalool	0.84 (±0.02)	3.25 (±0.06)
Carvacrol	-	4.51 (±0.05)
β-Caryophyllene	-	1.14 (±0.03)

## Data Availability

Not applicable.

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
