# Peer review of "Porous Coatings to Control Release Rates of Essential Oils to Generate an Atmosphere with Botanical Actives"

_materials, 2022, doi:10.3390/ma15062155_

Round 1

Reviewer 1 Report

The findings in the manuscript entitled "Porous coatings to control release rates of essential oils to generate an atmosphere with botanical actives" is interesting and would be beneficial for the application of EOs. The following are comments about the manuscript:

  1. Please proofread the manuscript. There are typos and formatting errors such as on page 1, line 19 ('concentration', not 'concertation) and on page 2, line 83 (m2/g, not m2/g; cm3/g, not cm3/g). Also, use the multiplication symbol '×' and not the alphabet 'x' for page 4, line 114.
  2. Consistently use the abbreviation for FCC and EO in the manuscript. Do not mix the usage of the full terminology and the abbreviation. For examples, on page 2, line 80 (use 'FCC' instead of 'functionalized calcium carbonate (FCC)') and page 4, line 133 (use 'EOs' instead of 'essential oils').
  3. Move Figure 1 to the result section. Also, please label the FCC and PET on the image because this will help the readers to understand the description given in the text about the position of the FCC.
  4. Section 2.3. How did the authors make sure that the concentrations of EOs on the coating were 5, 10, 30 w/w %? This is because EOs are volatile, thus the concentration on the coating might be less than the originally prepared EOs.
  5. Section 2.4. Why did the temperature of the sample chamber set at 7°C and  23°C? Please include the reason for using these two temperatures in the methodology or the discussion section.
  6. Page 5, line 169. The 'c' on 'rc' should be subscript.
  7. Figure 2. Please change the line colour for the samples 10 wt% and 30 wt% because the difference is not that noticeable. Also, the insert in (b) is missing.
  8. Figure 4. Please explain in more detail the reason rosemary oil did not have a significant asymptotic mass loss at different temperatures.

Reviewer 2 Report

I congratulate the authors for conducting the present study.

The Title looks fine.

The Abstract should be conducted on a single paragraph or at least without spacing between paragraphs.

The keywords should be put in alphabetic order.

I recommend avoiding the bullet point paragraph on the Introduction.

The material and methods, as well as the results, seem clear.

I recommend the authors to debate the study strenght, limitations and future recommendations.

Please review the reference list, some references are not according to the journal guidelines. 

Round 2

Reviewer 2 Report

Dear author, I have no more concerns. Thank you.